# *Citrus* By-Products: Valuable Source of Bioactive Compounds for Food Applications

**DOI:** 10.3390/antiox12010038

**Published:** 2022-12-25

**Authors:** Mariana A. Andrade, Cássia H. Barbosa, Muhammad Ajmal Shah, Nazir Ahmad, Fernanda Vilarinho, Khaoula Khwaldia, Ana Sanches Silva, Fernando Ramos

**Affiliations:** 1Department of Food and Nutrition, National Institute of Health Doutor Ricardo Jorge, Av. Padre Cruz, 1649-016 Lisbon, Portugal; 2Faculty of Pharmacy, University of Coimbra, Coimbra, Azinhaga de Santa Comba, 3000-548 Coimbra, Portugal; 3REQUIMTE/LAQV, Rua D. Manuel II, Apartado 55142, 4051-401 Oporto, Portugal; 4MEtRICs, Departamento de Ciências e Tecnologia da Biomassa, Departamento de Química, NOVA School of Science and Technology, Universidade NOVA de Lisboa, FCT NOVA, Campus de Caparica, 2829-516 Caparica, Portugal; 5Department of Pharmacy, Hazara University, Mansehra 21120, Pakistan; 6Department of Pharmacology and Clinical Pharmacy, Faculty of Pharmacy, Universitas Gadjah Mada, Sekip Utara, Yogyakarta 55281, Indonesia; 7Laboratoire des Substances Naturelles, Institut National de Recherche et d’Analyse Physico-Chimique, INRAP, Pôle Technologique de Sidi Thabet, Tunis 2020, Tunisia; 8National Institute for Agricultural and Veterinary Research (INIAV), I.P., Rua dos Lagidos, Lugar da Madalena, 4485-655 Vairão, Portugal; 9Center for Study in Animal Science (CECA), ICETA, University of Oporto, 4051-401 Oporto, Portugal; 10Associate Laboratory for Animal and Veterinary Sciences (AL4AnimalS), 1300-477 Lisbon, Portugal

**Keywords:** active food packaging, antioxidant and antimicrobial activities, citrus by-products, essential oils, phenolic compounds

## Abstract

*Citrus* production produces about 15 million tons of by-products/waste worldwide every year. Due to their high content of bioactive compounds, several extraction techniques can be applied to obtain extracts rich in valuable compounds and further application into food applications. Distillation and solvent extraction continues to be the most used and applied extraction techniques, followed by newer techniques such as microwave-assisted extraction and pulsed electric field extraction. Although the composition of these extracts and essential oils directly depends on the edaphoclimatic conditions to which the fruit/plant was exposed, the main active compounds are D-limonene, carotenoids, and carbohydrates. Pectin, one of the most abundant carbohydrates present in *Citrus* peels, can be used as a biodegradable polymer to develop new food packaging, and the extracted bioactive compounds can be easily added directly or indirectly to foods to increase their shelf-life. One of the applications is their incorporation in active food packaging for microbiological and/or oxidation inhibition, prolonging foods’ shelf-life and, consequently, contributing to reducing food spoilage. This review highlights some of the most used and effective extraction techniques and the application of the obtained essential oils and extracts directly or indirectly (through active packaging) to foods.

## 1. Introduction

*Citrus* fruits are one of the most produced crops in the world. According to FAOSTAT data [1], its world production in 2020 was 158,490,986 tons, an increase of approximately 7.5% compared to 2017, being oranges the most produced *Citrus* fruit (Table 1). Around 30 million tons are used for juice production from this kind of citrus [2]. Juices, jams, concentrated formulas, pastes, and other fruit formulations can easily originate tons of fruits by-products targeted to several cattle feed, bioethanol production, or the extraction of compounds with powerful antimicrobial and antioxidant activities. However, these by-products still originate tons of waste which, due to their high content in bioactive compounds, must be discarded in a responsible and eco-friendly way which, in consequence, can increase the final cost of the final product [3,4]. Every year, it is estimated that 15 million tons of Citrus by-products/waste are produced, worldwide [5].

*Citrus* is a genus belonging to the Rutaceae family of trees and shrubs, including oranges, lemons, grapefruits, tangerines, and limes [6]. The origin of the *Citrus* genus is a topic of debate in the scientific community since some scientists defend that the origin of the genus was in Southeast Asia and other scientists in Australia [7].

*Citrus* fruits’ by-products can be divided into peels (flavedo and albedo), seeds, and pulp residue. The flavedo is the outside colorful part of the peel that contains the oil sacs and, the albedo is the white interior part of the peel, rich in pectin [7]. Peels are also rich in sugars and have a high concentration of D-limonene, a powerful antimicrobial compound [2,8,9]. Pulp residue is constituted by the segment wall or membranes that contain the juice [10]. Seeds are mainly composed of nitrogen-free extract, lipids, crude protein, and fiber [10]. These by-products have several active compounds with powerful bioactive activities that have a very important role in the food industry, such as antioxidant and antimicrobial capacities.

As well as every other fruit, the chemical composition of *Citrus* fruits and their by-products, varies with the edaphoclimatic conditions that the plant is exposed to. These fruits are known for their high content of vitamin C and carotenoids with pro-vitamin A, namely β-cryptoxanthin (Table 2) [11,12]. Carotenoids are natural compounds responsible for the yellow, orange, and red colors in fruits and plants, helping the plants in photosynthesis and defense against light oxidation [13,14]. These compounds are very important for human health, acting against carcinogenesis, and preventing cardiovascular and degenerative diseases, so their presence is of utmost importance in the human diet [15]. Regarding vitamin C, *Citrus* fruits can contribute to vitamin intake with 40% of the Dietary Reference Intake (DRI) [16].

Consumers’ demand for more ‘natural’ and high-quality products is increasing, representing an important challenge for the food industry. Generally, the food industries resort to synthetic food additives, which are economically viable, chemically stable, and easily applied, to improve or maintain foods’ quality and prolong their shelf-life. However, there has been an increasing concern about the use and direct application of some of these compounds since they have been associated with allergic reactions, the promotion of carcinogenesis, and the appearance of neurodegenerative diseases [3,25,26,27].

In this line, the main purpose of this review is to critically discuss the potential of *Citrus* by-products based on their bioactive compounds and their applicability to foods as direct or indirect additives.

## 2. Active Compounds of *Citrus* By-Products

### 2.1. Bioactive Compounds Extraction

As stated before, the chemical composition of *Citrus* fruits and their by-products are directly dependent on the edaphoclimatic conditions to which the plant is exposed. Additionally, the chemical composition varies, varies mainly according to the species and cultivar. However, in terms of its essential oils (EOs) and extracts, the extraction protocol directly influences the content of the active compounds and the yield. The most common extraction method to obtain EOs and extracts are hydro and steam distillation, solvent extraction, and cold pressing [28,29,30]. “Greener” and recent extraction techniques, such as microwave extraction, ultrasound extraction, and supercritical fluid extraction, have emerged as a more sustainable alternative to more traditional techniques since they require less use of energy and solvent(s) [31,32]. The application of *Citrus* EOs is extensive and transversal to several industries, including the food, cosmetic, and pharmaceutical industries [33,34].

Steam distillation is the most used method to obtain EOs from plants. Generally, using this method for EOs extraction, around 93% yield is obtained. Briefly, the application of heat in the form of steam is responsible for the breakdown of the cell structure of the plant material and the consequent release of the essential oil [35,36,37,38]. Sikdar & Baruah [39] compared the extracted essential oils from orange, sweet lime, and lemon peels obtained with steam distillation varying the applied temperature and extraction time. For all the conditions, essential oil from orange peels presented the highest yield, followed by the sweet lime essential oil. The authors reach the optimal conditions of 96 °C for a period of 60 min [39]. Hydrodistillation is often used to extract essential oils from flowers and wood. This technique consists of the complete immersion of the plant material in water, followed by heating the mixture until boiling and the condensation of the steam and essential oil vapor to an aqueous phase. The water protects the oil from overheating, acting as a barrier [38]. Although a relatively economical and easy-to-apply technique, the extraction by distillation processes presents some disadvantages such as low efficiency, loss of volatile compounds, long extraction times, and degradation of unsaturated or ester compounds as a result of the use of high temperatures [38].

Solvent extraction is a conventional technique used mostly for extracting compounds from fragile parts of plants, such as flowers. Usually, contrary to distillation techniques, it does not resort to high temperatures, protecting the active compounds that are thermo-sensitive. Several solvents can be used in this extraction, such as hexane, ethanol, methanol, and acetone. However, the solvent choice is directly dependent on the final use of the extract or essential oil, since it is possible that the toxic solvent may be present in the extract or essential oil [38]. The solvents and the authorized additives to be directly or indirectly used are specified in the European Commission Regulation No 231/2012 and its amendments [40].

To overcome some limitations of the solvent extraction techniques, researchers started to apply supercritical fluids as solvents in this technique. A supercritical fluid is any substance at a pressure and temperature above its endpoint of a phase equilibrium curve (critical point), below the pressure required to compress it into a solid, where there is no distinction between the gas and liquid phase [41]. The extraction by supercritical fluids presents a higher efficiency and a lower loss of volatile compounds than the previous extraction methods. Supercritical carbon dioxide is one of the most used fluids in this type of extraction. Besides being eco-friendly, the use of carbon dioxide allows the extraction process to occur at relatively low temperatures since its critical temperature is 31 °C, as well as its easy application at high-pressure conditions, presents in a liquid form [38,42]. However, this extraction technique using carbon dioxide has disadvantages, due to its non-polar properties. Although, this can be compensated by adding other solvents such as ethanol, methanol, and water [42,43,44]. Menichini et al. [45] compared the essential oil extracted from *Citrus medica* L. cv. Diamante peels by three different methods: hydrodistillation, supercritical CO_2,_ and cold pressing. Limonene was the major compound found in the essential oils extracted by hydrodistillation and cold pressing followed by γ-Terpinene, while in the essential oil obtained with the supercritical CO_2_ extraction, the major compound was citropen (84.5%), followed by 2,3-Dihydrobenzofuran (2.9%) [45]. Also, the authors found that the essential oil obtained by supercritical CO2 presented no anti-inflammatory activity, while the essential oils obtained by hydrodistillation and cold-pressing presented anti-inflammatory activity [45]. Sicari & Poiana [46] compared the EOs extracted through hydrodistillation, solvent extraction by Soxhlet with pentane, and supercritical CO_2_ extraction from kumquat (*Fortunella margarita* Swingle) peels. All three essential oils presented almost the same content in limonene (around 96%) and their chemical composition was not significantly different. However, the EO obtained with supercritical CO_2_ presented a slightly higher content in esters and sesquiterpenes, which improved the essential oil aroma [46].

Having emerged in the 20th century, microwave extraction, or microwave-assisted extraction (MAE), is one of the most applied extraction techniques. Microwaves, located between the higher infrared frequencies and the lower radio frequencies, are non-ionizing electromagnetic waves [47]. In the MAE, microwaves act as energy vectors which, when applied to a certain material will absorb the electromagnetic energy and transforms it into heat [48,49]. The transformation of electromagnetic energy into heat relies on two mechanisms, that can occur simultaneously in both the sample and the solvent: ionic conduction and dipole rotation [48,50]. This guarantees that the system heating takes place at the same time, meaning, that the heating of both the solvent and the solid matrix occurs at the same time, unlike other extraction techniques where the heating occurs from the outside to the inside of the matrix and the mass transference occurs from the inside to the outside [50]. When compared with the more conventional/traditional extraction techniques, MAE presents several advantages, such as the use of lower quantities of solvent and lower human exposure to the used solvent, significant reduction in the extraction time, higher selectivity of the extracted compounds and the possibility of a solvent-free extraction [48,50,51,52]. However, not all are advantages regarding MAE. Method optimization is one of them. Several parameters must be considered when implementing/developing an MAE method, such as applied power, extraction time, solvent: matrix ratio, and matrix composition [48]. The choice of solvent is particularly important. Although both polar and non-polar solvents can be used, the choice must consider the solvent’s dielectric properties: a low dissipation factor translates into less dissipated heat, originating from the absorption of the microwave energy [48,50]. For instance, the water has a very low dissipation factor, which can lead to superheating and the extraction of some thermo-sensitive compounds is not advised [48]. Ferhat et al. [53] compared the extraction of EOs from fresh lemon (*Citrus limon* L.) peels by microwave accelerated distillation (or microwave ‘dry’ distillation) with the conventional techniques of cold pressing and hydrodistillation. The microwave extraction resulted in a higher yield with a lower extraction time period. Also, the oxygenated fraction in the OE extracted with microwaves was 10% higher than the essential oil extracted with hydrodistillation and 40% higher than the OE extracted by cold pressing [53]. Bustamante et al. [54] also compared MAE of EOs from orange peels with hydrodistillation extraction, stating that MAE EO possessed slightly higher quantities of monoterpenes (0.78% higher), including D-Limonene, α-pinene, β-pinene and γ-Terpinene [54].

Usually applied to liquid and semi-solid foods, Pulsed Electric Field (PEF) extraction is one of the most recent extraction techniques applied in the food industry. Usually applied to liquid and semi-solid foods, consists of applying short pulses, micro- or milliseconds, of high voltage between 10 to 80 kV/cm, to the food placed between two electrodes [55,56]. The application of short high-voltage pulses increases the cell membrane conductivity and permeability due to the incensement of the transmembrane potential [56,57]. PEF is largely applied in the food industry to assure food microbiological safety since it has the advantage of inactivating pathogenic microorganisms without having to apply high temperatures, maintaining the original sensorial (texture, flavor, color) and nutritional value of unprocessed foods [55]. Coupled with other extraction techniques, such as solvent extraction, PEF can be used as a tool to improve the extraction or recovery of valuable compounds, such as phytochemicals. For instance, Hwang et al. [58] applied PEF to subcritical water extraction in *Citrus unshiu* peels improving the hesperidin content from 38.45 mg/g to 46.96 mg/g. Also, Kantar et al. [59] applied PEF in the extraction of polyphenols with ethanol extraction from orange pomelo and lemon. The authors found that the application of the PEF treatment increased the polyphenol content of ethanolic extracts by 50%. In addition, it can also increase the efficiency of juice extraction and increase the yield, from fruits by-products and plants, of bioactive compounds, extracts, and essential oils [55,57]. Luengo et al. [18] used PEF by applying 1, 3, 5 and 7 kV/cm to sweet orange (*C. sinensis*) peels, increasing the orange peels’ antioxidant capacity extract by 51%, 94%, 148%, and 192%, respectively. The authors also concluded that the total polyphenol extraction yield increased by 20%, 129%, 153%, and 159% for the respective applied high voltages and, for the extract obtained with the 5 kV/cm, the content of naringin from 1 to 3.1 mg/100 g of fresh weight (FW) of peel and hesperidin from 1.3 to 4.6 mg/100 g FW of peel [18]. In another study led by El Kantar et al. [59]), PEF was applied to orange, pomelo, and lemon fruits in aqueous media at 3 kV/cm. The authors found that the applied current increased the juice yield by 25% for oranges, 37% for pomelo, and 59% for lemons [59]. In a more recent study, led by Peiró et al. [21], an electric field of 7 kV/cm was applied to lemon peels, which increased the polyphenol extraction by 300%, with astonishing contents of hesperidin (84 mg/100 g FW) and eriocitrin (176 mg/100 FW).

### 2.2. Active Compounds of Citrus Fruits By-Products

*Citrus* fruits and their by-products present a large spectrum of phytochemical compounds. Phenolic compounds (namely flavonoids), terpenoids, carotenoids, vitamins, fatty acids, and aromatic compounds are among them [6,8,60]. Some of the active compounds that can be found in extracts and essential oils from *Citrus* fruits by-products can be observed in Table 2, as well as the chromatographic methods used for their determination.

Beyond the edaphoclimatic conditions, the composition of the extracts and EOs obtained from *Citrus* by-products is also dependent on the processing of the fruit itself (for example, to obtain the juice) and the extraction method applied to the by-products (temperature conditions, solvent, time of the extraction, among others) [60].

Peels are the major by-product of *Citrus* fruits’ industrial processing and are responsible for most of the commercialized *Citrus* EOs. The oil composition may vary but, approximately, 90% is composed of *D*-limonene [8]. Linalool, β-myrcene, and α-pinene can also be found in high amounts [5,22,61,62]. Flavonoids are another class of compounds that can be easily found in *Citrus* fruits by-products, being neoeriocitrin, neohesperidin and naringin the main flavanones in lemon (*Citrus limon*), orange (*Citrus aurantium*) and bergamot (*Citrus bergamia* Fantastico) peels [17].

*Citrus* fruits are rich in carotenoids, specially α-carotene, β-carotene, lutein, zeaxanthin, and β-cryptoxanthin [63]. Carotenoids are well-known for their antioxidant activity, and their moderated consumption is related to the reduction of the incidence of cancer, arteriosclerosis, and arthritis and the promotion of immune functions and 0.1 to 0.5% of the dry weight of *Citrus* peels is composed by carotenoids [63,64]. β-cryptoxanthin can be easily found in oranges, tangerines, and mandarins, representing a very important role in human nutrition since it has a powerful antioxidant capacity and provitamin A activity [65].

Limonene (D-limonene, _L_-limonene) is the major compound present in *Citrus* by-product extracts and EOs. Is a monocyclic monoterpene with low toxicity (oral LD 50 values 5–6 g/kg), registered as Generally Recognized as Safe (GRAS) in the Code of Federal Regulations (CFR) for its use as synthetic flavoring [66,67]. Limonene is widely used in the pharmaceutical, food, and cosmetic industries as a fragrance in perfumes, soaps, and household cleaning products. Also, it can be found in some pesticides and insect repellents [68]. Limonene has several clinical applications and is recognized for its anticancer, anti-asthmatic, and anti-microbial activities [66,67]. Due to its ability to dissolve cholesterol, D-limonene has been clinically used to dissolve gallstones containing cholesterol. It is also used to neutralize heartburn due to its action on gastric acid [67,68].

Although not considered an active compound or a phytochemical, pectin is a major compound in *Citrus* peels, representing, generally, 20–30% of the dry weight of peels [63]. Present in the peels (flavedo and albedo), central column and juice sac of *Citrus* fruits, limes, lemons, grapefruits, and oranges are the fruits with a higher pectin content. It is highly used in the food industry as a thickener and stabilizer for jams and juices [63]. Pectin enhances gastric motility and nutrient absorption, and has shown preventative and therapeutical effects on cancer, diabetes, high blood pressure, and obesity [64]. The pharmaceutical industry uses pectin in the production of plasma and hemostatic agents and laxatives [63]. Barbosa et al. [19] extracted polyphenols from industrial *Citrus* juice by-products (*Citrus latifolia* and four cultivars from *Citrus sinensis*) and the remaining residue from the pectin extraction of those by-products. The authors found that, in total, the extract from the juice by-products presented a lower content of polyphenols than the extract from the pectin by-products. Extract from the pectin by-products presented a higher content of hesperidin (314.44 mg/100 g DM vs.232.65 mg/100 g DM), naringin (3.11 mg/100 g DM vs.1.02 mg/100 g DM), and tangeretin (6.07 mg/100 g DM vs.1.41 mg/100 g DM). However, the extract from the juice by-products obtained a higher content of narirutin (29.34 mg/100 g DM vs.17.50 mg/100 g DM) and ellagic acid (10.97 mg/100 g DM vs.0.27 mg/100 g DM). Also, the authors found that the antioxidant activity of the juice by-product extract was higher than the antioxidant activity of the pectin by-product extract [19].

### 2.3. Biological Activity of Citrus Fruits By-Products

As described earlier, *Citrus* fruits and their by-products are known for their health benefits. These benefits are due to their biological activities, like antioxidant, anticarcinogenic, anti-tumor, antimicrobial, and anti-inflammatory properties. EOs, the main product of the extraction of *Citrus* peels, have powerful biological activities, especially antimicrobial potential [30]. *Citrus* EOs are used for their germicidal, antioxidant, and anticarcinogenic properties [28].

In the study led by Han et al. [69], 57 participants were exposed to bergamot essential oil, which improved the participants’ positive feelings by 17% compared with the control group. Another study, led by Matsumoto et al. [70], proved that the Japanese citrus fruit yuzu (*Citrus junos* Tanaka) EO has an anti-stress effect and eases premenstrual emotional symptoms, namely tension–anxiety, anger–hostility, and fatigue—common. Mazloomi et al. [71] concluded that orange seed protein concentrate could reduce blood pressure and help diabetes management. Menezes Barbosa et al. [72] proved the antimicrobial activity of *Citrus* by-products from juice and pectin extraction of *Citrus latifolia* and four cultivars of *Citrus sinensis* (‘Hamlin’, ‘Valência’, ‘Pêra Rio’, and ‘Pêra Natal’), against *Bacillus cereus*, *Staphylococcus aureus*, *Listeria monocytogenes*, *Escherichia coli*, and *Salmonella* Typhimurium. Ruviaro et al. [24] research showed the vasorelaxation potential of hesperetin, a flavanone commonly present in *Citrus* by-product extracts. The authors also improved the extraction of this compound by resorting to enzyme-assisted extraction of pectin *Citrus* by-products (Table 2). Yue Liu et al. [23] observed the antifungal activity of *Citrus reticulata* (Mandarin) peel ethanolic extract against *Aspergillus flavus*, a major producer of aflatoxins which present a serious health risk.

## 3. Direct Application of Citrus By-Products, Their EOs, and Extracts to Foods

### 3.1. Prolonging Foods’ Shelf-Life

Due to their high nutrient and phytochemical content, *Citrus* by-products are used in several ways, for different and distinct purposes. Lately, studies have been made to evaluate the benefits of the application of *Citrus* by-products and their extracts and EOs to foods [73,74,75].

Microbial growth in foods is one of the major concerns in the food industry. Usually, the manufacturers resort to several antimicrobial compounds to inhibit those microorganisms and prolong foods’ shelf-life. These additives are regulated in the European Union through Regulation No 1333/2008 and all of its amendments [76]. Benzoic acid and its derivatives (E210–219, E928, and E1519), nitrates and nitrites (E240–E259), and sorbic acid and its derivatives (E200 and E202) are the most commonly used antimicrobial additives. However, their safety for human consumption has been brought to question, so finding safe substitutes has become a priority [25,77,78,79,80,81,82,83,84,85,86,87,88]. Lipid oxidation is, also, one of the major concerns of the food industry, being one of the major causes of food spoilage [89,90]. Antioxidant compounds can be used to stop or delay this chemical process, prolonging foods’ shelf-life and preventing the occurrence of off-flavors.

Fernández-López et al. [91] studied the ability of lemon and orange extracts to inhibit bacterial growth and to extend the storage shelf-life of cooked meatballs, as well as their antioxidant activity. Although both extracts showed significantly low malonaldehyde (MDA) values, indicating antioxidant activity, meatballs with orange extracts showed lower MDA values than those with lemon extracts at the end of 12 days of storage. Also, during the storage time, in the meatballs with *Citrus* extracts, lactic acid bacteria were not detected, which suggested that the extracts could be more effective to control lactic acid bacteria growth during storage time [91]. This could be due to the high fiber content extracts, which have high water absorption, and, consequently, reduce microbial growth [91,92,93]. Devatkal, Narsaiah, & Borah [94] and Devatkal & Naveena [95] studied the antioxidant properties of kinnow rind powder extracts. They used it as a natural antioxidant instead of a synthetic one, in goat meat. In both cases, the use of the extract was successful as the lipid oxidation was significantly reduced, during refrigerated storage. This was corroborated by the low MDA values after refrigerated storage [94,95].

Spinelli et al. [96] enriched the nutritional quality of fish burgers with a micro-encapsulated extract from orange epicarp and then evaluate the bio-accessibility of phenolic, flavonoid, and carotenoid compounds. They observe an increase in the bio-accessibility of the bioactive compounds, concluding that enriching fish burgers with a micro-encapsulated extract from orange epicarp is beneficial since it increases the quality of the food [96].

Bambeni et al. [97] applied an extract obtained from orange (*C. reticulata*) pomace to beef patties and compared the lipid oxidation and the microbial growth with beef patties with no treatment and with beef patties treated with synthetic additive sodium metabisulphite (SMB). Although the beef patties with SMB presented lower MDA and an inferior microbiological growth than the patties treated with the orange extract, SMB contains sulfites which have been associated with the occurrence of asthma and allergic responses [97,98,99]. However, it is noteworthy that the orange extract presented lower MDA content and lower microbiological growth than the control patties, showing an antioxidant and antimicrobial effect [97].

Tayengwa et al. [100] feed Angus steers with dried *Citrus* pulp, consisting of comprised seeds, pulp, and peels. The authors found that the α-tocopherol content of the Angus feed with the citrus pulp was three times higher than the control group and the MDA content of the beef was also significantly lower than the control group. Regarding the antimicrobial analysis, the group fed with the citrus pulp showed a reduction in coliforms than the control group [100]. This study shows the importance of the active compounds’ biological activities and reinforces the use of *Citrus* by-products in the animal and food industry, reinforcing the existence of a circular economy and waste management regarding the fruit industry. Following this line of thought, Wu et al. [101] investigated the antifungal potential of golden finger citron (*Citrus medica* L. var. sarcodactylis) flowers, fruits, and leaves EO in Chinese steamed bread. The EO obtained from the leaves prolonged the Chinese steam bread shelf-life for longer periods (11 to 13 days) than the EOs from the flowers (4 to 5 days) and fruits (3 to 5 days). The authors also observed a significantly higher antifungal activity of the leave EO than the antifungal activity of the synthetic preservative potassium sorbate [101].

Shehata et al. [102] evaluated the potential of an orange peel extract in inhibiting the lipid oxidation of vegetable oil and compared it to the synthetic additive butylated hydroxytoluene (BHT). The authors found that the vegetable oil with the orange peel extract presented lower peroxide values than the control and the oil with BHT, showing that the orange peel extract is more effective against the oils’ lipid oxidation than the synthetic additive [102]. Nishad et al. [103] incorporated grounded goat meat with a citrus peel extract and evaluated its lipid oxidation after three and six months of storage. The authors found that the meat with the citrus peel extract exhibits significantly lower MDA values and peroxide values than the control meat [103].

### 3.2. Foods’ Quality Improvement through CITRUS By-Products

This section indicates the quality improvement of foods by using Citrus phytochemicals, such as texture and color. Additionally, to the phytochemicals with powerful biological activities, *Citrus* by-products are also a great source of dietary fiber. This dietary fiber is preferable to other sources, such as cereals, due to their high content of bioactive compounds [92,93,104]. Dietary fiber, known for its water and fat-biding properties, is widely used in meat and meat products to improve cooking yield and texture [6,29,60,92,93,104,105]. Fernandez-Gines et al. [105] studied the influence of the addition of *Citrus* by-product fiber and the storage condition of bologna sausage. They manufactured the bologna sausage with different concentrations of citrus fiber (0.5, 1, 1.5, and 2%). The bologna sausages with added *Citrus* fiber showed a significant decrease in residual nitrite level. The addition of *Citrus* fiber, significantly, altered the color parameters and the textural characteristics. Specifically, lightness values were increased on the sausage with *Citrus* fiber, but no differences were found between fiber concentrations. On the other hand, aspect, saltiness, fatness, residual taste, and pH levels were not significantly affected by *Citrus* fiber addition. Also, no microbial growth was observed in the sausage with *Citrus* fiber. MDA values were higher in the sausage stored under lighting conditions when compared with those stored under darkness, for all citrus fiber concentrations [105]. Fernández-López et al. [106] also studied the benefits of fiber by incorporating *Citrus* by-products, specifically lemon albedo, and orange dietary fiber powder, into cooked and dry-cured sausage. The study concluded that, in both cases, the nitrite levels produced were significantly lower. Furthermore, the color parameters were altered and TBA values were higher in the sausage stored under lighting conditions than those stored under darkness [106].

Pectin is a very common dietary fiber obtained from *Citrus* peel. Pectin is mostly used as a thickener, stabilizer, and emulsifier. Thus, it is used to produce jams, jellies, marmalade, fruit juice, confectionary products, and bakery fillings. It is also used for the stabilization of acidified milk drinks and yogurts [29,60].

There are several examples of the direct application of *Citrus* by-products EOs and extracts and their antimicrobial and antioxidant potential. However, similar to the concerns with synthetic additives, the safety ingestion limits and their long-term effects on human health are still unknown and there are several variables to be explored. Nevertheless, active food packaging can be a suitable short-term solution for the reduction of the concentration of synthetic additives and the application of natural additives.

## 4. Application of Citrus By-Products to Active Food Packaging

With technological advances, new concepts and materials began to emerge. From a traditional/conventional perspective, the main purpose of packaging is the protection of foods from external factors without interacting with the food’s matrix. In an attempt to overcome the shortcomings of conventional food packaging, intelligent and active food packaging has emerged. Regarding the active packaging systems, their objective is to directly interact with the packaged food to extend food shelf-life. According to the European Legislation, an active package can “change the composition or the organoleptic properties of the food only if the changes comply with the Community provisions applicable to food, such as the provisions of Directive 89/107/EEC (4) on food additives” [107]. There are two kinds of active packaging: absorbent packaging and releasing packaging. The first type is designed to interact with foods absorbing compounds from the packaged food or the headspace of the package, without having the active substance(s) or component(s) migrate to foods. The most common, within this type, in the market are moisture and oxygen absorbents. Regarding the releasing packages, the polymeric matrix is loaded with active compounds that will migrate gradually into the packaged food to increase food’s shelf life, through the delay of the phenomena responsible for food deterioration such as inhibition of microorganisms and/or lipid oxidation. These packages can also be used for maintaining, enhancing, or improving food’s organoleptic characteristics [108].

The most common material used in food packaging is plastic, which raises an enormous environmental concern. Several biopolymers are being proposed to replace conventional plastics due to the massive environmental concern raised by non-biodegradable plastics and plastics obtained from non-renewable resources. For instance, Kraft paper is widely used for packaging but it has several disadvantages such as high permeability to gas and moisture [109], which can be improved with *Citrus* by-products and their extracts and EOs. Kasaai & Moosavi [110] successfully enhanced the water and gas barrier of Kraft paper using mandarin peel and hydrophobic leaf extracts.

Due to their high content of carbohydrates, *Citrus* by-products are being used for the production of bacterial cellulose (BC) and polyhydroxyalkanoate (PHA) [111]. BC, described for the first time in 1988, is a polymer of β-1,4-linked glucose produced by aerobic bacteria [112,113]. BC has several interesting properties such as high crystallinity, high cellulose purity, high tensile strength, and water-holding capacity. It is used in the food industry to produce fruit cocktails and jellies [114]. Mostly, the BC is produced using coconut water as a medium for the bacteria (generally *Gluconacetobacter xylinum*), which is a limited source since coconuts only grow in tropical areas. To overcome this problem, Cao et al. [114] resorted to *Citrus* pulp water, resulting from the juice extraction of *Citrus* fruits, to produce BC, and compared the production of BC through coconut water. The authors concluded that the BC grown in the *Citrus* pulp water almost reached industrial levels. The medium promoted the growth of BC with different physicochemical features (higher water-holding capacity and low hardness) [114]. Güzel & Akpınar [115] produced BC from *Komagataeibacter hansenii* GA2016, using peels from lemon, mandarin, orange, and grapefruit, with a yield between 2.06 to 3.92% with a higher water holding capacity than the BC usually produced, high crystallinity and thermal stability, with a thin fiber diameter.

Arrieta et al. [116] studied the influence of the incorporation of *D*-limonene in PLA and poly-hydroxybutyrate (PHB), a biodegradable thermoplastic obtained from microorganisms under physiological stress. The authors manage to obtain five different formulations. As expected, the PLA presented a colorless and transparent appearance, while the PHB presented an amber color with light transparency. Visually, regarding the films’ transparency and color, there were no apparent differences between the films without D-limonene and the films with *D*-limonene. In conclusion, the film with PLA:PHB, on a ratio of 75:25, incorporated with *D*-limonene, can offer transparent and flexible films, with water-resistant properties and enhanced oxygen barrier, suitable for biodegradable food applications [116].

Muñoz-Labrador et al. [117] transformed industrial pectin obtained from *Citrus* fruits by-products (lemons and limes peels) in coatings that were able to increase the quality of strawberries for 5 days, when compared with stored strawberries with no treatment.

Wu et al. [118] resorted to pomelo peels to produce a biodegradable film, using the dried and ground pomelo peels with sodium alginate and glycerol. The authors also incorporated tea polyphenols, rich in catechins, to increase the antimicrobial and antioxidant properties of the active film. The films were applied to soybean oil for a maximum storage time period of 30 days. The authors obtained flexible and transparent films suitable for oil packaging. The incorporation of the tea polyphenols increases the film’s antioxidant and antimicrobial activity against Escherichia coli and *Staphylococcus aureus* [118]. In a study led by Kaanin-Boudraa et al. [119], Citrus × paradisi extract was obtained through MAE with 40% ethanol, and incorporated in a multilayer LDPE-PET active packaging. The multilayer film with 10% of extract presented the highest antioxidant activity, followed by the film with 5% extract [119].

Li et al. [120] successfully developed a new packaging film made from pectin from orange peels, sodium alginate, and pterostilbene, with low water vapor permeability, good barrier properties, and antioxidant activity. Pectin extracted from citrus by-products-based film incorporated with green propolis extract, with antioxidant activity, was also developed by Marangoni Júnior et al. [121].

Nanoparticles can be used in food packaging to reinforce the polymeric matrix. Gao et al. [122] compared commercial ZnO nanoparticles with ZnO nanoparticles synthesized with *Citrus sinensis* peel extract. They also incorporated the nanoparticles in carboxymethylcellulose (CMC) to obtain a coating to be applied to strawberries. The authors found that the ZnO synthesized with the *Citrus* extract presented a higher antimicrobial activity, similar cytotoxicity, and similar crystallinity when compared with the commercial ZnO nanoparticles [122].

Yanjie Li et al. [120] incorporated orange peels EO, by casting method, into fish (*Cynoglossus semilaevis*) skin gelatin and chitosan, at different percentages (0.25, 0.5, and 1.0%, *v*/*v*). The authors observed that the addition of the EO increase the films’ thickness from 43.29 μm (in the control film) to 86.95 μm (in the film with 1.0% of EO). The EO addition decreased the water vapor permeability to 0.86 × 10^–11^ g m^−1^ s^−1^ Pa^−1^ (in the film with 0.5% of EO) and increased the elongation at break and films’ opacity. Also, the antioxidant activity increased with the continuous addition of the orange EO, from 14.80% DPPH free radical inhibition and 4.67% ABTS free radical inhibition, to 49.38% and 57.71%, respectively. The active films also presented antimicrobial activity against Escherichia coli and Staphylococcus aureus [120].

In another research, carried out by Roy and Rhim [123], grapefruit seed extract was incorporated into a poly(vinyl alcohol) to form an active film. The addition of the grapefruit seed extract increased the films’ thickness from 72.9 to 75.1 μm, the tensile strength from 28.6 MPa to 31.1 MPa, the water vapor permeability from 4.18 × 10^–10^ g·m/m^2^·Pa·s to 4.43 × 10^–10^ g·m/m^2^·Pa·s, and the elongation at break from 148.0% to 158.0%. However, the elongation modulus decreased from 0.68 GPa to 0.44 GPa. Regarding the films’ biological activities, the addition of the grapefruit seed extract significantly increased the antioxidant percentage in the DPPH free radical scavenging assay from 0.7 to 50.3% and in the ABTS free radical scavenging assay, from 2.9% to 90.2%. The active film also presented antimicrobial activity against *E. coli* and a remarkable antimicrobial activity against *L. monocytogenes* [123].

Evangelho et al. [124] incorporated *Citrus sinensis* peels EO, at different quantities (0.3, 0.5, and 0.7 μL/g), in corn starch films by casting and evaluating its antimicrobial activity and its properties. All films showed antimicrobial activity against *S. aureus* and *L. monocytogenes*. This antimicrobial activity increases with the increase of the EO content. The films’ thickness and opacity decreased with the addition of the EO, being the active film with 0.5 μL/g of EO the most thicken, with 0.142 μm, and the active film with 0.7 μL/g of EO the opaquest (16.24%). The films’ tensile strength decreased with the addition of the EO, from 5.11 MPa (control film) to 2.40 MPa (film with 0.7 μL/g) [124].

EOs and extracts, as well as other compounds extracted from *Citrus* by-products, can be used as the active additive in the food packaging polymeric matrix to prevent or delay food spoilage, but also, can be used to extract pectin or to produce BC or BHA as a substitute of the conventional polymeric matrix of the food package itself.

## 5. Conclusions and Future Perspectives

With *Citrus* fruits being one of the most consumed fruits in the world, they are also one of the most significant sources of food waste among fruits. These by-products are rich in a wide variety of active components which could be applied in several industries for numerous purposes. With technological advances, new and more efficient extraction methods, such as microwave-assisted extraction or pulsed electric field extraction, using less aggressive/toxic solvents and less energy, obtaining higher yields without compromising the extracts and EOs quality. However, most of the studies evaluate the individual by-products or by-products produced on a small scale. Therefore, there it is necessary to better characterize industrial Citrus by-products and to standardize its EOs and extracts to guarantee their quality and effectiveness.

*Citrus* by-products have enormous industrial potential, from polymers for plastic-like food coatings and packaging to active compounds with antioxidant and antimicrobial activities. For instance, pectin can be the base polymer for a new form of food packaging or coating, being a possible substitute for plastic. *Citrus* by-products can also be used for nanoparticle stabilization and used as a direct or indirect food additive. Nevertheless, the toxicity of these bioactive compounds remains unknown and their use in food must be extensively studied as well as the future effects on human and environmental health.

## Figures and Tables

**Table 1 antioxidants-12-00038-t001:** Production of Citrus fruits in 2019 according to the Food and Agriculture Organization of the United Nations (FAO) [1].

Fruit	Harvested Area (ha)	Production (Tonnes)
Oranges [common, sweet orange (*Citrus sinensis*); bitter orange (*C. aurantium*)]	4,060,129	78,699,604
Tangerines, mandarins, clementines, satsumas [mandarin, tangerine (*Citrus reticulata*); clementine, satsuma (*C. unshiu*)]	2,756,887	35,444,080
Lemons and limes [lemon (*Citrus limon*); sour lime (*C. aurantifolia*); sweet lime (*C. limetta*)]	1,226,617	20,049,630
Citrus Fruits [Some minor varieties of citrus are used primarily in the preparation of perfumes and soft drinks, including bergamot (*Citrus bergamia*); citron (*C. medica* var. cedrata); chinotto (*C. myrtifolia*); kumquat (*Fortunella japonica*)]	1,508,639	14,496,484
Grapefruit (inc. pomelos) [*Citrus maxima*; *C. grandis*; *C. paradisi*]	346,191	9,289,462
**Total**	9,898,463	157,979,260

**Table 2 antioxidants-12-00038-t002:** Chromatographic techniques for the determination and quantification of some bioactive compounds found in *Citrus* by-product extracts.

Species/ Variety	Common/ Local Name	Main Bioactive Compounds and Levels Found	Chromatographic Technique/ Apparatus	Chromatographic Method	Ref.
Peel from *Citrus microcarpa*	Kumquat	Quercetin (0.78 ± 0.003 mg/g, db) β-cryptoxanthin (37.0 ± 1.45 μg/g, db) Lutein (36.4 ± 1.56 μg/g, db) Zeaxanthin (36.4 ± 1.57 μg/g, db) Caffeic acid (17.3 ± 1.57 μg/g, db) β-carotene (2.79 ± 0.14 μg/g, db)	Reversed-phase HPLC with UV detector. Column: LiChrospher^®^100 RP18e, 5 µm, 4.0 mm internal diameter × 250 mm	**Flavonoids**: MPA: 2% acetic acid (aqueous) MPB: 0.5% acetic acid (aqueous)-acetonitrile (*v*/*v*; 50:50) Flow rate: 1 mL/min. Gradient: 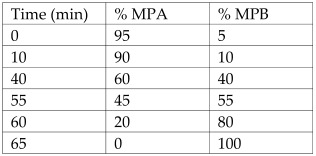 **Carotenoids**: MPA: Acetonitrile MPB: Methanol MPC: Dichloromethane All mobile phases contained 0.1% BHT, 0.1% triethylamine, and 0.005 M ammonium acetate (in methanol). Gradient: 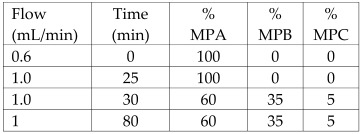	[17]
Peel from *Citrus reticulata* × *Citrus sinensis*	Murcott	Diosmin (0.40 ± 0.01 mg/g, db) Chlorogenic acid (339 ± 4.01 μg/g, db) Sinapic acid (178 ± 5.62 μg/g, db) Zeaxanthin (25.2 ± 0.99 μg/g, db) β-cryptoxanthin (16.9 ± 0.75 μg/g, db) Lutein (13.3 ± 0.51 μg/g, db) β-carotene (12.1 ± 0.51 μg/g, db)			
Peel from *Citrus sinensis* (L.) Osbeck	Liucheng	Naringin (23.9 ± 0.32 mg/g, db) Hesperidin (20.7 ± 0.38 mg/g, db) Sinensetin (0.42 ± 0.01, mg/g, db) β-carotene (50.2 ± 2.28 μg/g, db) Lutein (29.3 ± 1.17 μg/g, db) Zeaxanthin (27.7 ± 1.21 μg/g, db) β-cryptoxanthin (0.76 ± 0.04 μg/g, db)			
Peel from *Citrus grandis* Osbeck CV	Peiyou	Naringin (29.8 ± 0.20 mg/g, db) Caffeic acid (27.5 ± 1.74 μg/g, db)			
Peel from *Citrus tankan* Hayata	Tonkan	Hesperidin (23.4 ± 0.25 mg/g, db) β-carotene (36.9 ± 1.38 μg/g, db) Zeaxanthin (11.6 ± 0.58 μg/g, db)			
Peel from *Citrus reticulata* Blanco	Ponkan	Hesperidin (29.5 ± 0.32 mg/g, db) Kaempferol (0.38 ± 0.002 mg/g, db) Rutin (0.29 ± 0.004 mg/g, db) Luteolin (0.21 ± 0.01 mg/g, db) *p*-Coumaric acid (346 ± 2.45 μg/g, db) Ferulic acid (150 ± 4.89 μg/g, db) β-carotene (69.2 ± 2.67 μg/g, db) β-cryptoxanthin (30.5 ± 1.26 μg/g, db)			
Peel from *Citrus limon* (L.) Bur	Lemon	Rutin (0.29 ± 0.002 mg/g, db) Caffeic acid (80.0 ± 3.72 μg/g, db) β-carotene (10.3 ± 0.47 μg/g, db) Lutein (2.95 ± 0.12 μg/g, db) Zeaxanthin (0.81 ± 0.04 μg/g, db) β-cryptoxanthin (0.81 ± 0.04 μg/g, db)			
*Citrus sinensis* peels	Sweet orange	Naringin (3.1 mg/100 g FW) Hesperidin (4.6 mg/100 g FW)	HPLC-PDA Column: reversed-phase column Microsorb-MV 100-5 C18 (25 × 0.46 cm; 5 μm particle size) with a pre-column (5 × 0.46 cm; 5 μm particle size) of the same material	Column temperature: 40 °C MPA: water-formic acid solution (95:5) MPB: ACN Injection vol: 10 µL Flow rate: 1 mL/min Gradient: 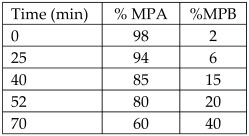	[18]
Juice by-products from *Citrus latifolia* and four cultivars from *Citrus sinensis*	-	Hesperidin (232.65 ± 10.47 mg/100 g DM) Narirutin (29.34 ± 0.43 mg/100 g DM) Ellagic acid (10.97 ± 0.08 mg/100 g DM) Tangeretin (1.41 ± 0.04 mg/100 g DM) Hesperetin (1.05 ± 0.04 mg/100 g DM) Naringin (1.02 ± 0.05 mg/100 g DM)	HPLC-DAD Column: C-18 Acclaim 120 column (Dionex, 3 µm, 4.6 × 150 mm)	Column temperature: 30 °C MPA: 0.1% of formic acid in water MPB: 0.1% of formic acid in methanol Flow rate: 0.6 mL/min Gradient: 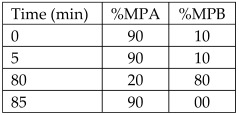	[19]
*Citrus* fruits juice by-products (non-specified)	*Citrus* fruits juice by-products (non-specified)	Narirutin (9.6 ± 1.5 mg/100 g DM) Hesperidin (99.7 ± 7.4 g/100 g DM) Naringenin (22.6 ± 0.6 g/100 g DM) Hesperetin (80.8 ± 13.7 g/100 g DM) Tangeritin (1.7 ± 0.2 g/100 DM)	HPLC with a Dionex UltiMate 3000 chromatography system Column: C18 Acclaim^®^ 120 column (Dionex, 3 μm, 4.6 × 150 mm)	Column temperature: 30 °C Detector: UV/VIS detector (DAD-3000) Wavelength: 280 nm MPA: water-formic acid (99.9:0.1, *v*/*v*) MPB: methanol-formic acid (99.9:0.1, *v*/*v*) Flow rate: 0.6 mL/min	[20]
		Narirutin (8.1 ± 0.9 g/100 g DM) Hesperidin (88.3 ± 5.8 g/100 g DM) Naringenin (21.1 ± 2.6 g/100 g DM) Hesperetin (82.5 ± 11.9 g/100 g DM) Tangeritin (1.6 ± 0.1 g/100 g DM)			
		Narirutin (50.9 ± 4.5 mg/100 g DM) Hesperidin (228.9 ± 7.0 mg/100 g DM) Tangeritin (1.1 ± 0.1 mg/100 g DM)			
		Narirutin (27.1 ± 0.2 mg/100 g DM) Hesperidin (117.3 ± 1.6 mg/100 g DM) Naringenin (19.5 ± 0.6 mg/100 g DM) Hesperetin (51.8 ± 2.1 mg/100 g DM) Tangeritin (1.3 ± 0.1 mg/100 g DM)			
Peels from lemon	-	Hesperidin (84.44 ± 8.35 mg/100 g FW) Eriocitrin (176.35 ± 15.39 mg/100 g FW)	HPLC with a diode-array detector Column: C18 reverse phase column	Column temperature: 40 °C MPA: Acidified bidistillated water (0.1% of glacial acetic acid) MPB: Acidified acetonitrile (0.1% of glacial acetic acid) Flow rate: 0.5 mL/min Gradient: 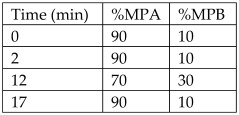	[21]
Peel from *Citrus limon*	Algerian oranges	Limonene (94.427%) β-Myrcene (2.158%) Linalool (0.293%) Valencene (0.165%) Octanal (0.435%)	GC/MS coupled with mass spectrometry (HP 6890 (II) interfaced with an HP 5973 mass spectrometer) Column: capillary column RTX-5 MS (30 m, ID 0.25 mm, film thickness 0.25 lm)	Carrier gas: Helium Flow rate: 1 mL/min Temperature ramp: 40 °C for 8 min; increased to 180 °C at 3 °C/min; increased to 230 °C at 20 °C/min	[22]
*Citrus reticulata* peels	Mandarin	Narirutin (2044.46 ± 55.48 µg/g of Sample) Hesperidin (1346.44 ± 67.78 µg/g of Sample) Nobiletin (218.02 ± 7.29 µg/g of Sample) Rutin (214.50 ± 9.28 µg/g of Sample) Taxifolin (134.36 ± 3.71 µg/g of Sample) Sinensetin (113.82 ± 4.73 µg/g of Sample)	HPLC coupled with DAD Column: Agilent Eclipse XDB-C18 (4.6 × 250 mm, 5 µm) reverse phase column	Column temperature: 40 °C Injection volume: 10 µL MPA: Acidified water (0.5% of formic acid) MPB: Acetonitrile Flow rate: 0.5 mL/min Gradient: 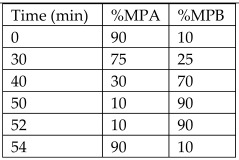	[23]
Crude orange juice by-products extract	Orange	Hesperidin (25.62 ± 0.44 mg/g LE) Narirutin (3.96 ± 0.10 mg/g LE)	HPLC coupled with DAD Column: Acclaim^®^ 120 C18 column (Dionex, 3 μm, 4.6 × 150 mm)	Column temperature: 30 °C MPA: Acidified water (0.1% of formic acid) MPB: Acidified methanol (0.1% of formic acid) Flow rate: 0.6 mL/min	[24]
Enzyme-treated orange juice by-products extract	Orange	Hesperetin (22.02 ± 0.48 mg/g LE) Hesperidin (6.08 ± 0.09 mg/g LE) Naringenin (2.23 ± 0.01 mg/g LE)			
Crude orange pectin by-products extract	Orange	Hesperidin (81.17 ± 3.01 mg/g LE) Narirutin (6.73 ± 0.21 mg/g LE) Tangerintin (1.28 ± 0.08 mg/g LE)			
Enzyme-treated orange pectin by-products extract	Orange	Hesperetin (43.70 ± 0.79 mg/g LE) Hesperidin (11.11 ± 0.39 mg/g LE) Naringenin (3.49 ± 0.10 mg/g LE)			

**Legend**: MPA—Mobile Phase A; MPB—Mobile Phase B; MPC—Mobile Phase C; HPLC—High-performance liquid chromatography; DAD—Diode Array Detector; PDA—Photodiode-Array Detection; GC—Gas Chromatography; MS—Mass Spectrometry; DM—Dry matter; DW—Dry weight; FW—Fresh weight; db—dried base; LE—Lyophilized Extract.

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
