# Peer review of "Citrus By-Products: Valuable Source of Bioactive Compounds for Food Applications"

_antioxidants, 2022, doi:10.3390/antiox12010038_

Round 1
Reviewer 1 Report
Comments:
Lines 82-88- Are all the synthetic additives a concern? I would rephase this sentence to ‘’ some of these compounds’’.
Lines 183-184- Can you include numerical results in % as you did before?
Table 2- For consistency, in ref. 54 and 56, I would include the gradient elution in a table as you did with the other references. Also, in ref. 59 and 57 the gradient is not described. Please complete.
Lines 304-306- Please include more reference to support this statement.
Line 361- Can you be more specific about the type of improvement? Nutritional improvement?
Reviewer 2 Report
in the abstract line 24 avoid posterior, use further or following
in section 2.
The sentence in the first two lines does not consider other factor influencing the chemical composition as well as species and cultivars before edaphoclimatic conditions
line 102: when and where steam distillation obtains 93% of yield or what ? The sentence is not clear
regarding the use of PEF on line 190-191 the electroporation mechanism should be explained since the PEF application has to be coupled with a solvent or hydro extraction since the PEF process is used as pre-treatment. It is not itself an extraction process
in section 4, the paragraph 4.1 is not useful for the purpose of the paper.
the sentence in line 442 is too general regarding the plastic materials and the environmental issue
In the whole manuscript there is small attention to the differentiation among Citrus products, generally referring to the general term “citrus” where probably some differences could be found in lemon, orange, ecc. production and processing as well as in compounds used to valorize byproducts or wastes
Reviewer 3 Report
Please, sort keywords alphabetically
Table 2: flow rate: 1 mL/min
